# External validation of the improving partial risk adjustment in surgery (PRAIS-2) model for 30-day mortality after paediatric cardiac surgery

Lucia Cocomello [1] Massimo Caputo,[2] Rosie Cornish [3] Deborah Lawlor[4]

[1]MRC Integrative Epidemiology Unit, University of Bristol, Bristol, UK
[2]Bristol Heart Institute, University of Bristol, Bristol, UK
[3]Population Health Science, Bristol Medical School University of Bristol, Bristol, United Kingdom
[4]MRC Integrative Epidemiology Unit, Department of Social Medicine, University of Bristol, Bristol, UK

**Correspondence to**
Dr Lucia Cocomello;
nn18747@bristol.ac.uk

## ABSTRACT

**Objective** Independent temporal external validation of the improving partial risk adjustment in surgery model (PRAIS-2) to predict 30-day mortality in patients undergoing paediatric cardiac surgery.

**Design** Retrospective analysis of prospectively collected data.

**Setting** Paediatric cardiac surgery.

**Intervention** PRAIS-2 validation was carried out using a two temporally different single centre (Bristol, UK) cohorts: Cohort 1 surgery undertaken from April 2004 to March 2009 and Cohort 2 from April 2015 to July 2019. For each subject PRAIS-2 score was calculated according to the original formula.

**Participants** A total of 1352 (2004-2009) and 1197 (2015-2019) paediatric cardiac surgical procedures were included in the Cohort 1 and Cohort 2, respectively (median age at the procedure 6.3 and 7.1 months).

**Primary and secondary outcome measures** PRAIS-2 performance was assessed in terms of discrimination by means of ROC (receiver operating characteristic) curve analysis and calibration by using the calibration belt method.

**Results** PRAIS-2 score showed excellent discrimination for both cohorts (AUC 0.72 (95%CI: 0.65 to 0.80) and 0.88 (95%CI: 0.82 to 0.93), respectively). While PRAIS-2 was only marginally calibrated in Cohort 1, with a tendency to underestimate risk in lowrisk and overestimate risk in high risk procedures (P-value = 0.033), validation in Cohort 2 showed good calibration with the 95% confidence belt containing the bisector for predicted mortality (P-value = 0.143). We also observed good prediction accuracy in the non-elective procedures (N = 483;AUC 0.78 (95%CI 0.68 to 0.87); Calibration belt containing the bisector (P-value=0.589).

**Conclusions** In a single centre UK-based cohort, PRAIS-2 showed excellent discrimination and calibration in predicting 30-day mortality in paediatric cardiac surgery including in those undergoing non-elective procedures. Our results support a wider adoption of PRAIS-2 score in the clinical practice.

## INTRODUCTION

Congenital heart diseases (CHDs) are the most common birth defect affecting between 6 and 8 per 1000 of live born children in

### Strengths and limitations of this study

► A strength of the present study is that data were prospectively collected as part of the UK National Congenital Heart Disease Audit, and as such, they have undergone continuous and inclusive systematic validation that includes the review of a sample of case notes by external auditors to ensure coding accuracy.

► We used a recently proposed method (calibration belt) that does not require patients to be categorised according to risk percentile but rather provides a risk function across all risk value with 95% CI providing a measure of uncertainty/imprecision.

► A key limitation of this study is that the sample size is relatively small and considerably smaller than the cohort used to develop PRAIS-2.

middle-income and high-income countries.[1] [2] Around 5000 paediatric cardiac surgical procedures are performed each year in the UK. Despite overall mortality being low (3%), there is notable variation in mortality rates between centres. Mortality rates, and variation between centres, are monitored by public and healthcare regulatory bodies as part of assessing the quality of care provided by the centres. In order to monitor a centre's quality of care and performance and make fair comparisons between centres in terms of mortality following paediatric cardiac surgery, it is essential to take account of differences in case mix across centres.[3] This requires accurate risk stratification. However, CHD includes a large spectrum of diagnoses with a wide range of surgical procedures performed in the context of a relatively small number of patients. These characteristics make risk stratification extremely challenging.

Several methods that aim to provide an objective risk assessment for taking account of case mix when assessing centre performance have been developed. These include

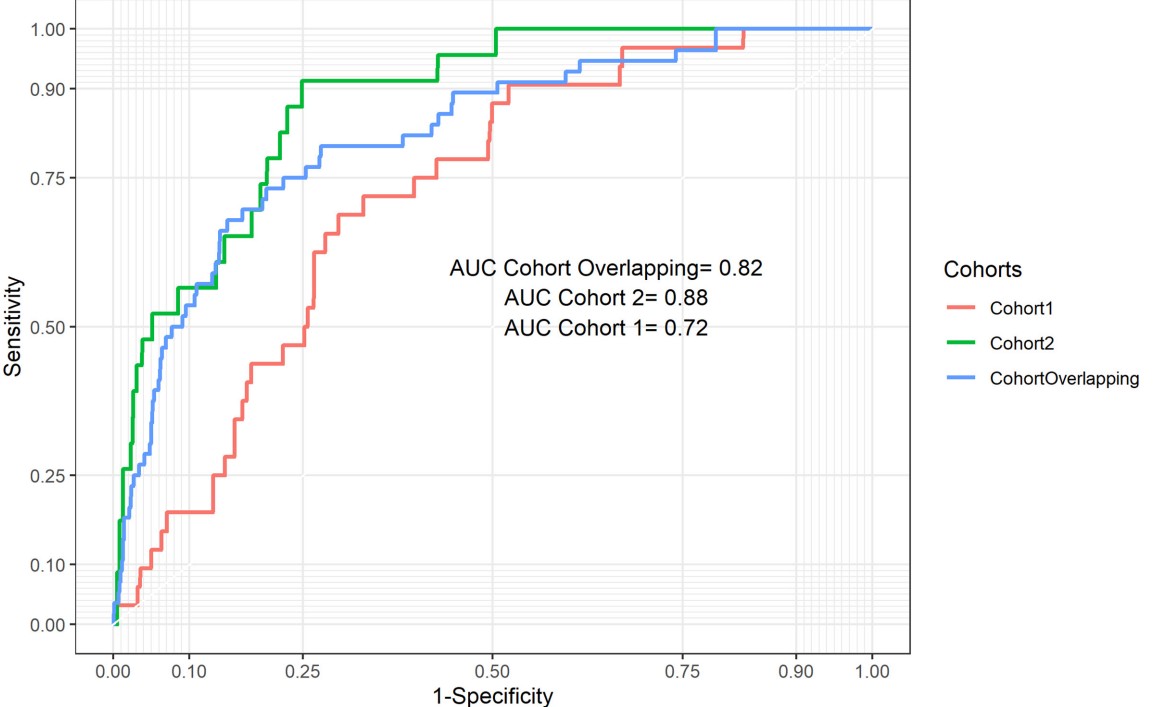

**Figure 1** ROC (receiver operating characteristic curve) curve for PRAIS-2 in the independent Cohort 1, 2 and Overlapping Cohort.

consensus-based methods, such as the Risk Adjustment for Congenital Heart Surgery-1 Categories (RACHS-1 Categories)[4] and Aristotle,[5] and more recently empirical research-based methods, such as The Society of Thoracic Surgeons–European Association for Cardio-Thoracic Surgery Congenital Heart Surgery Mortality Categories (STAT Mortality Categories),[6] the Society of Thoracic Surgeons Congenital Heart Surgery Database (STS-CHSD) Mortality Risk Model for Congenital Cardiac Surgery,[7–9] and the partial risk adjustment in surgery (PRAIS),[10] which has been developed in the UK and proposed for measuring between centre variation in mortality across the UK. The most recently updated version, PRAIS-2,[11] was developed to predict 30-day mortality using data from the UK National Congenital Heart Disease Audit (NCHDA).[12] In comparison to PRAIS, PRAIS-2 included more detailed information about acuity, diagnosis and comorbidities and was shown to have better discrimination and calibration than the original PRAIS.[13] However, adoption of PRAIS-2 into clinical practice is still limited and this has been partially attributed to the lack of external independent validation other than the one presented by the authors in the original paper.[14] That validation used data from the same UK dataset but was undertaken on a temporally independent subcohort (figure 1). While such validation within a study is important, using the same cohort (and same group of investigators) can result in risk of optimism bias, which would tend to exaggerate risk stratification accuracy in comparison to findings from independent cohorts and investigators.[15 16] Moreover, prediction models can present significant calibration drift due to temporal and geographic differences in case mix, patients characteristics and surgical technique; the impact of this on the performance of PRAIS-2 is unknown.[17 18] Lastly, as the data used to develop PRAIS-2 did not include information on whether the procedures were elective or conducted as non-elective (emergencies, urgency and salvage), it was not possible to determine whether performance was similar in both of these situations. It is not implausible that stratification accuracy will differ between the two.

The purpose of this study was to perform an external independent validation of the PRAIS-2 score in a cohort from a single tertiary paediatric UK centre (in Bristol, South West England). This single UK centre contributed to the cohort in which PRAIS-2 was originally developed, but with data from different time periods than used here. In this study, we determine performance discrimination separately in two Bristol Heart Institute cohorts: (1) procedures performed earlier than those used in the initial development and validation of PRAIS-2 (Cohort 1) and (2) procedures performed after those used in the initial development and validation of PRAIS-2 (Cohort 2). To explore possible calibration drift, we have compared calibration of the model prediction between three Bristol Heart Institute cohorts; procedures undertaken 2004–2009 (Cohort 1), 2009–2015 (Cohort overlapping the dataset included in PRAIS-2 development) and 2015–2019 (Cohort 2). In a subsample, we were also able to undertake the first (exploratory) analysis of how well PRAIS-2 performs in those undergoing non-elective procedures.

**Table 1** Flowchart of procedures included in the original PRAIS-2 validation and the current study

| | PRAIS-2 national sample | Bristol cohort |
|---|---|---|
| April 2004–March 2009 | None | n=1352 Fully independent validation Cohort 1 and exploration of temporal calibration drift |
| April 2009–March 2014 | n=21 838 Original training | n=1515 Geographical and temporal calibration drift, Overlapping Cohort |
| April 2014–March 2015 | n=4207 Original external validation | n=309 Combined with 1515 above for geographical and temporal calibration drift, Overlapping Cohort |
| April 2015–July 2019 | None | n=1197 Fully independent validation Cohort 2 and exploration of temporal calibration drift (with priority data) |
| | Elective n=714 | Non-elective=483 First exploration of whether PRAIS-2 accurately stratifies non-elective procedures |

## METHODS

The study complies with the Declaration of Helsinki. As this analysis came under clinical audit/quality of care assessment and all data were anonymised following the governance criteria of the NHS, the institutional review board agreed informed consent was not required.

### Patient and public involvement

This research was done without patient and public involvement.

### Data source

Data were obtained from the Bristol dataset which is part of the NCHDA within the National Institute of Cardiovascular Outcomes Research (NICOR). As such, the data used here have undergone continuous and inclusive systematic validation, which includes the review of a sample of case notes by external auditors to ensure coding accuracy.[12] Table 1 summarises the data sources used in this study and their relationship to the UK National data used in the development of PRAIS-2.

The entire (ie, including all three cohorts used in any analyses) Bristol data consisted of 4886 paediatric cardiac surgical procedures (defined as surgery on the heart or great vessel in patients aged <16 years old, excluding catheter procedures and trivial/minor procedures) performed between April 2004 and July 2019. Procedures with missing information on mortality (133), one or more of the variables used in the calculation of PRAIS-2 calculation (371) or both (9) were excluded. Procedures with missing information on one or more of the variables used to calculate PRAIS-2 showed a higher rate of 30-day mortality (online supplemental table S1). For those with missing data on one or more of the variables used to calculate PRAIS-2, the variables that were available suggested that they had higher risk profiles than those with no missing variables, both in the whole study data (online supplemental table S1) and when analysed in the three separate temporal cohorts (online supplemental tables S2–S4). For example, in those with at least one missing variable (n=371), the proportion with severe illness was 22.9%, whereas in those with no missing variables (n=4373) it was 7%. The same results in Cohort 1 were 15.5% and 0%, in the overlapping with PRAIS-2 discovery were 0% and 3.3% and in Cohort 2 were 18.5% and 55.6%. The remaining 4373 (90% of the 4886 eligible) procedures were included in the analysis. PRAIS-2 was developed using national data (including that from the Bristol cohort) from patients undergoing procedures between April 2009 and March 2015. In these analyses, we use Bristol data from April 2004 to March 2009 (Cohort 1) and April 2015 to July 2019 (Cohort 2) as independent external validation. While we acknowledge that treatments and mortality rates have changed since 2004, and the first validation set may not reflect contemporary practice, we would expect this to primarily affect model calibration; finding good discrimination for this earlier set would support external validity and generalisability of PRAIS-2. Seeing good discrimination across all time periods would provide valuable evidence about the generalisability of the PRAIS-2 model. In addition, we explored discrimination and calibration in the Bristol cohort that was also included in the national cohort to develop PRAIS-2, April 2009 to March 2015 (Overlapping Cohort). This allows us to explore any evidence of geographical (centre) difference in stratification performance and using all three cohorts to explore the extent of temporal calibration drift.

### PRAIS-2 prediction score and its constituent variables

PRAIS-2 score is generated from a transformed logistic regression model of 30-day mortality following cardiac surgery.[13] The model included the following perioperative variables: age, weight, diagnosis, procedure group, type of procedure, whether or not there was definite univentricular heart function, additional cardiac risk factors, acquired comorbidity, congenital comorbidity, severity of illness and an additional coefficient for procedures performed after 2013 (online supplemental table S1 shows the units or categories of each of these variables). The formula for the PRAIS-2 score (using the function from the logistic regression model) is

$$\frac{1}{1+e^{-z}}$$

where z is the logistic model function of the nine variables. In all analyses, we used the PRAIS-2 model (equation) that was recalibrated by the original authors after their internal validation (online supplemental equation 1).

## Outcome

The primary outcome was 30-day all-cause mortality. Information on mortality was obtained from the Office for National Statistics (ONS).

## Statistical analysis

Continuous data were presented as medians with IQR. Categorical variables were presented as counts with percentages. We assessed the discriminative ability of the PRIAS-2 model to distinguish between children who did and did not die within 30 days using the area under the receiver operating characteristic curve (AUC) analysis[19] and its calibration by comparing observed to predicted 30-day mortality rates for groups of procedures with different predicted levels of risk.[14 20 21] Model performance in the main analysis (Cohort 1 and Cohort 2) was also evaluated by reporting sensitivity, specificity, positive-likelihood and negative-likelihood ratios (LRP and LRN) and the positive and negative predictive values (PPV and NPV) using cut-off values that included 10%, 20%, 80% and 90% above the cut-off.[22]

Overall model calibration was assessed using the calibration belt method. This method is based on a generalisation of Cox's regression modelling where the relation between the logits of the probability predicted by a model and of the event rates observed in a sample is represented by a polynomial function whose coefficients are fitted and its degree is fixed by a series of likelihood-ratio tests.[23] Risk underestimation is suggested if the calibration belt with its 95% CI is above the bisector (perfect prediction line), while overestimation is indicated by the belt with its 95% CI being below the bisector. The calibration belt method was accompanied by the Hosmer-Lemeshow $\chi^2$ goodness-of-fit test.[24] Comparison of $\chi^2$ were used to investigate the extent of calibration drift across different eras. All analyses were performed using R V.3.5.2 and the following packages: givitiR, ROCit, ggplot2, ResourceSelection.

## RESULTS

The final study population consisted of 1352 procedures in Cohort 1 (median age (IQR) 6.3 (1.4–30.8)), 1197 procedures in Cohort 2 (median age (IQR) 7.1 (2.2–51.0)), which also included 483 non-elective procedures, and 1824 procedures in the Overlapping Cohort (median age (IQR) 5.9 (1.3–38.6)).

Table 2 shows procedures characteristics of the two external validation (Cohort 1 and Cohort 2) and the subgroup, from Cohort 2, who had non-elective procedures, and the Overlapping Cohort. While the average PRAIS-2 score did not differ across the temporally different

cohorts, we noticed a downward shift in mortality rate in the most recent Cohort 2.

The predicted risk in the external validation cohorts (Cohort 1 and Cohort 2), the Overlapping Cohort and the non-elective subset were 1.1%, 1.0%, 1.1.% and 2.9%, respectively, and the observed mortality was 2.4%, 1.9%, 3.1% and 4.6%, respectively (online supplemental figure 1).

In the Cohort 2 external validation analysis, the PRAIS-2 score showed good discrimination ability (AUC 0.87; 95% CI 0.82 to 0.93, figure 1) and calibration between predicted and observed mortality across all risk categories with the 95% CI of calibration belt containing the bisector (p value=0.14 (figure 2; Hosmer and Lemeshow goodness-of-fit test $\chi^2$=8, df=8, p-value=0.5 (online supplemental table S6A). Sensitivity, specificity, LRP, LRN, PPV and NPB using different cut-off values of the prediction score are reported in table 3. Assuming safe effective interventions could reduce those identified at high risk of mortality, in this population we would want to correctly identify the majority of those at risk, that is, minimise false negative results by having a highly specific test, and we would want a positive test to have a high probability of identifying those most at risk (PPV). A 10% threshold maximises specificity (0.90) and maximised the PPV (although this is only 0.10). The 20% threshold also has high specificity (0.80) and a similar PPV (0.06) to the 10% threshold.

In Cohort 1, PRAIS-2 showed a lower but still good discrimination ability (AUC 0.72 (95% CI 0.65 to 0.80, figure 1), compared with Cohort 1, but was poorly calibrated, with a tendency to underestimate risk in low-risk procedures and overestimate in high-risk procedures (p value=0.033, figure 2; Hosmer and Lemeshow goodness-of-fit test $\chi^2$=19, df=8, p value=0.02, online supplemental table S6b). Online supplemental table S7 shows the discrimination metrics for this cohort, with results similar to those found for Cohort 2 and shown in table 2.

In additional analyses to check that data from Bristol were not notably different to the main UK-wide developmental cohort, we showed that the Overlapping Cohort also had good discrimination, with results similar to the whole developmental cohort (AUC 0.82 (95% CI 0.77 to 0.88, figure 1). However, the model was only marginally calibrated with a tendency to risk underestimation (figure 2). Hosmer and Lemeshow goodness-of-fit test $\chi^2$=16, df=8, p value=0.04 (online supplemental table S6c).

Considering calibration drift across all three temporally different cohorts, we observe modest improvement across all three cohorts such that the most recent cohort was the best calibrated. Specifically, the Hosmer-Lemeshow test $\chi^2$ decreased from 19 in Cohort 1, through 16 in the Overlapping Cohort to 8 in Cohort 2.

In the subgroup of non-elective procedures (a subgroup of Cohort 2 as these data were not available in the other two cohorts), the model showed good discrimination, with AUC 0.78 (95% CI 0.68 to 0.87, figure 3)

**Table 2**  Procedure characteristics in Cohort 1, Cohort 2, in the non-elective subset and Overlapping Cohort

| | Independent validation Cohort 1 (April 2004–March 2009) | Independent validation Cohort 2 (April 2015–July 2019) | Non-elective Cohort 2 subset (April 2015–July 2019) | Cohort Overlapping with PRAIS-2 cohort (April 2009–March 2015) |
|---|---|---|---|---|
| Total number of procedures | 1352 | 1197 | 483 | 1824 |
| Age, year (median, IQR) | 6.3 (1.4–30.8) | 7.1 (2.2–51.0) | 1.7 (0.3–5.0) | 5.9 (1.3–38.6) |
| Weight, kg (median, IQR) | 6.1 (3.4–12.5) | 6.9 (3.9–15.5) | 3.7 (3.1–5.3) | 6.0 (3.6–13.4) |
| Diagnoses group, n (%)* | | | | |
| GROUP 1 | 71 (5.3) | 107 (8.9) | 47 (9.7) | 154 (8.4) |
| GROUP 2 | 149 (11.0) | 163 (13.6) | 77 (15.9) | 194 (10.6) |
| GROUP 3 | 111 (8.2) | 103 (8.6) | 65 (13.5) | 125 (6.9) |
| GROUP 4 | 218 (16.1) | 232 (19.4) | 118 (24.4) | 222 (12.2) |
| GROUP 5 | 104 (7.7) | 122 (10.2) | 49 (10.1) | 123 (6.7) |
| GROUP 6 | 104 (7.7) | 137 (11.4) | 45 (9.3) | 100 (5.5) |
| GROUP 7 | 176 (13.0) | 127 (10.6) | 34 (7.0) | 312 (17.1) |
| GROUP 8 | 155 (11.5) | 105 (8.8) | 32 (6.6) | 192 (10.5) |
| GROUP 9 | 20 (1.5) | <5 (<1) | <5 (<1) | 25 (1.4) |
| GROUP 10 | 56 (4.1) | 14 (1.2) | 6 (1.2) | 95 (5.2) |
| GROUP 11 | 188 (13.9) | 85 (7.1) | 8 (1.7) | 282 (15.5) |
| Procedure group, n (%)* | | | | |
| GROUP 1 | <5 (<1) | 31 (2.6) | 22 (4.6) | 28 (1.5) |
| GROUP 2 | 29 (2.1) | 11 (0.9) | 10 (2.1) | 33 (1.8) |
| GROUP 3 | 107 (7.9) | 46 (3.8) | 34 (7.0) | 84 (4.6) |
| GROUP 4 | 99 (7.3) | 103 (8.6) | 73 (15.1) | 114 (6.2) |
| GROUP 5 | 239 (17.7) | 228 (19.0) | 126 (26.1) | 344 (18.9) |
| GROUP 6 | 148 (10.9) | 53 (4.4) | 36 (7.5) | 180 (9.9) |
| GROUP 7 | 21 (1.6) | 45 (3.8) | 17 (3.5) | 50 (2.7) |
| GROUP 8 | 86 (6.4) | 100 (8.4) | 12 (2.5) | 126 (6.9) |
| GROUP 9 | 20 (1.5) | 16 (1.3) | 5 (1.0) | 26 (1.4) |
| GROUP 10 | 36 (2.7) | 35 (2.9) | 7 (1.4) | 70 (3.8) |
| GROUP 11 | 36 (2.7) | 19 (1.6) | 8 (1.7) | 57 (3.1) |
| GROUP 12 | 32 (2.4) | 64 (5.3) | 1 (0.2) | 50 (2.7) |
| GROUP 13 | 87 (6.4) | 68 (5.7) | 5 (1.0) | 103 (5.6) |
| GROUP 14 | 38 (2.8) | 43 (3.6) | 7 (1.4) | 55 (3.0) |
| GROUP 15 | 217 (16.1) | 144 (12.0) | 11 (2.3) | 275 (15.1) |
| GROUP 20 | 156 (11.5) | 191 (16.0) | 109 (22.6) | 229 (12.6) |
| Bypass n (%) | 903 (66.8) | 911 (76.1) | 273 (56.5) | 1361 (74.6) |
| Univentricular heart category, n (%) | 143 (10.6) | 219 (18.3) | 95 (19.7) | 258 (14.1) |
| Severity of illness†, n (%) | 0 (0.0) | 222 (18.5) | 203 (42.0) | 86 (4.7) |
| Acquired comorbidity†, n (%) | 9 (0.7) | 207 (17.3) | 132 (27.3) | 94 (5.2) |
| Additional cardiac risk factors†, n (%) | 58 (4.3) | 82 (6.9) | 56 (11.6) | 52 (2.9) |
| Congenital comorbidity†, n (%) | 173 (12.8) | 343 (28.7) | 143 (29.6) | 274 (15.0) |
| PRAIS-2 (median, IQR) | 0.0115 (0.0037–0.0258) | 0.010 (0.004–0.027) | 0.029 (0.011–0.068) | 0.011 (0.003–0.027) |
| 30-day mortality, n (%) | 32 (2.4) | 23 (1.9) | 22 (4.6) | 56 (3.1) |

Continued

| | Independent validation Cohort 1 (April 2004–March 2009) | Independent validation Cohort 2 (April 2015–July 2019) | Non-elective Cohort 2 subset (April 2015–July 2019) | Cohort Overlapping with PRAIS-2 cohort (April 2009–March 2015) |
|---|---|---|---|---|

**Table 2** Continued

*As the diagnoses groups and procedure groups have lengthy text details, we have not provided them in the table; they can be found in online supplemental table 5.
†Exact number suppressed for disclosure control purposes.
‡Definitions of variables are given in online supplemental table 5.

and good calibration, with the bisector contained within the 95% confidence level of the calibration belt (p value 0.61; figure 4 and Hosmer and Lemeshow goodness-of-fit test, $\chi^2$=10, df=8, p value=0.3; online supplemental table S6d). We were not able to conduct the analysis restricted to elective procedures as all but one patients survived to 30 days following elective procedures.

## DISCUSSION

Risk prediction models play an important role in current paediatric cardiac surgical practice because they allow meaningful comparison of outcomes between institutions by adjusting for differing case mix. Moreover, they can also give useful information for surgical decision-making, preoperative patient information and quality assurance measures. Before implementing a risk model in clinical practice, it is important to confirm its prediction ability with external validation.[14 25]

To our knowledge, PRAIS-2 has never undergone external validation other than that originally presented by the authors who developed the model. It is important that prediction models are validated by independent researchers, as well as in independent samples, before being adopted into clinical practice because authors evaluating the performance of their own model may tend to be overly optimistic in interpreting results or selective reporting.[16 26] For this reason, we conducted an external validation study of PRAIS-2 in two cohorts from

the paediatric cardiac surgical procedures cohort at the Bristol Royal Hospital for Children. Our findings show that in these two cohorts, which are independent (of the original cohort used to develop PRAIS-2), the PRAIS-2 model has good discrimination and good calibration in the most recent (Cohort 2) cohort. That calibration was better in Cohort 2 than in the Overlapping Cohort was surprising as we might expect the cohort contributing to the development of PRAIS-2 to be the best calibrated. We speculate that the temporal improvement in calibration across all three cohorts from the same geographical region is likely to reflect changes over time in case mix, patient risk profile and improvement in perioperative care. While we saw an improvement in calibration over time, the amount of this was modest. However, a further validation of the PRAIS-2 in the coming years would be useful as marked drift could result in important levels of risk overestimation or underestimation.[18]

As PRAIS-2 was developed with data from across the UK and in our external validation we have only used data from Bristol, we wanted to make sure that prediction in Bristol was broadly consistent with the UK as a whole. We therefore explored the predictive accuracy of PRAIS-2 in the Bristol cohort that contributed to the original PRAIS-2 development and found good discrimination, but the model was only marginally calibrated with a tendency to underestimate risk. It is not uncommon for calibration to differ between cohorts from different regions or over

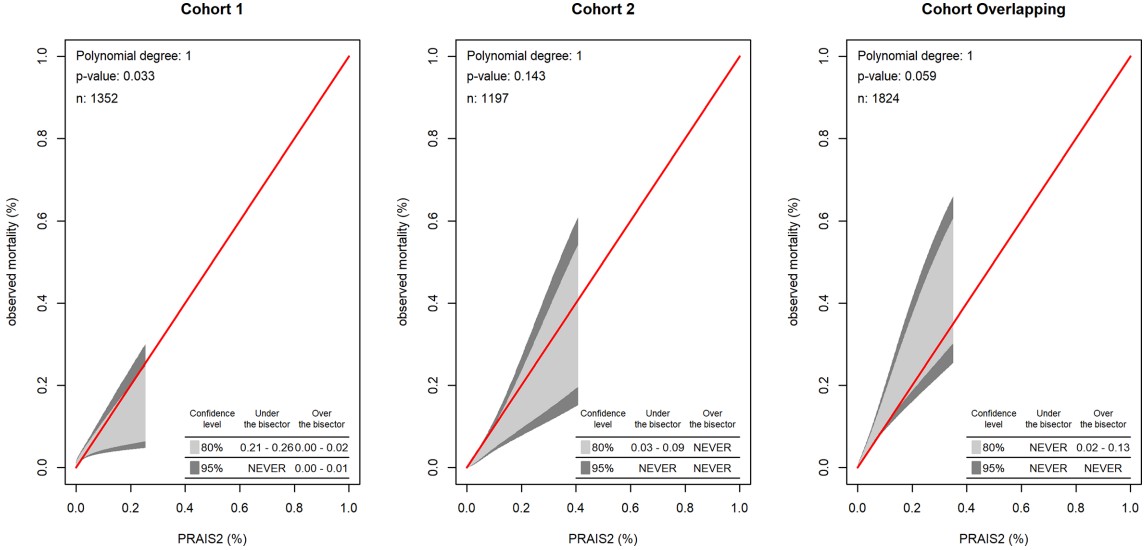

**Figure 2** Calibration belt of PRAIS-2 in the independent Cohort 1, Cohort 2 and Overlapping Cohort.

**Table 3** Metrics of model discrimination according to different PRAIS-2 cut-off values in Cohort 2

|          | Metrics | Top 90% | Top 80% | Top 20% | Top 10% |
|----------|---------|---------|---------|---------|---------|
| Cohort 2 | Cut-off | 0.001 | 0.003 | 0.03 | 0.07 |
|          | Sensitivity | 1 | 1 | 0.69 | 0.56 |
|          | Specificity | 0.10 | 0.20 | 0.80 | 0.90 |
|          | PPV | 0.02 | 0.02 | 0.06 | 0.10 |
|          | NPV | 1 | 1 | 0.99 | 0.99 |
|          | LRP | 1.11 | 1.25 | 3.64 | 6.20 |
|          | LRN | 0 | 0 | 0.37 | 0.47 |

LRN, negative-likelihood ratios; LRP, positive-likelihood ratios; NPV, negative predictive values; PPV, positive predictive values.

time. We are not able to determine variation in calibration between all of the different centres that contributed to the original PRAIS-2 development as data were not presented by centre in the original documentation of PRAIS-2 development.[13]

PRAIS-2 development excluded non-elective procedures and so its accuracy in this group, or whether this would be a valuable predictor, is unknown. We demonstrated that PRAIS-2 does have good discrimination and calibration in this higher risk subgroup. However, we acknowledge that our numbers were small for these analyses and this needs further exploration and validation in larger independent samples. We were not able to compare performance between those undergoing elective procedures and those undergoing non-elective

procedures as the mortality rate was very low, resulting in only one death among those undergoing elective procedures. This marked difference in mortality between elective and non-elective procedures suggests that elective versus non-elective status might be a valuable covariable in PRAIS-2, but large cohorts with these data would be needed to determine this.

A strength of the present study is that data were prospectively collected as part of the NCHDA national audit. As such, they have undergone continuous and inclusive systematic validation, which includes the review of a sample of case notes by external auditors to ensure coding accuracy.[12] A key limitation of this study is that the sample size is relatively small and considerably smaller than the cohort used to develop PRAIS-2, which included a total of 21 838 procedures (combining development and validation). Further, 10.5% of procedures had missing data on mortality and/or variables used to calculate PRAIS-2. These individuals tended to have higher risk profiles and their mortality rates were higher. Unlike the development cohort that included all UK patients, we only have data from one centre located in the South West of England where the population are more affluent and less ethnically mixed in comparison to the UK as a whole. Hence, further replication in a larger and more diverse population is necessary.

The Hosmer and Lemeshow goodness of fit test for calibration is widely used in assessing calibration,[27 28] but it provides a statistical test result comparing predicted to observed outcome rates across arbitrary grouping of deciles of risk, which may have limited clinical applicability. Therefore, we used a recently proposed method

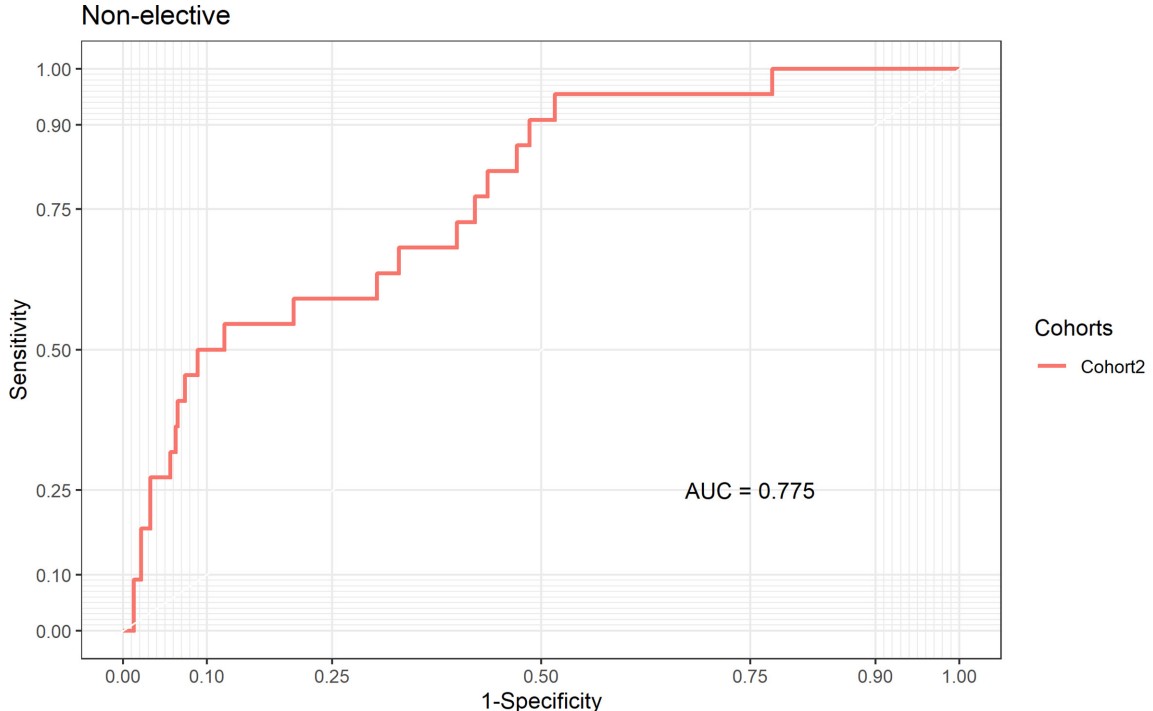

**Figure 3** Area under the receiver operating characteristic curve (AUC) in the subset of non-elective procedures in the external validation cohort.

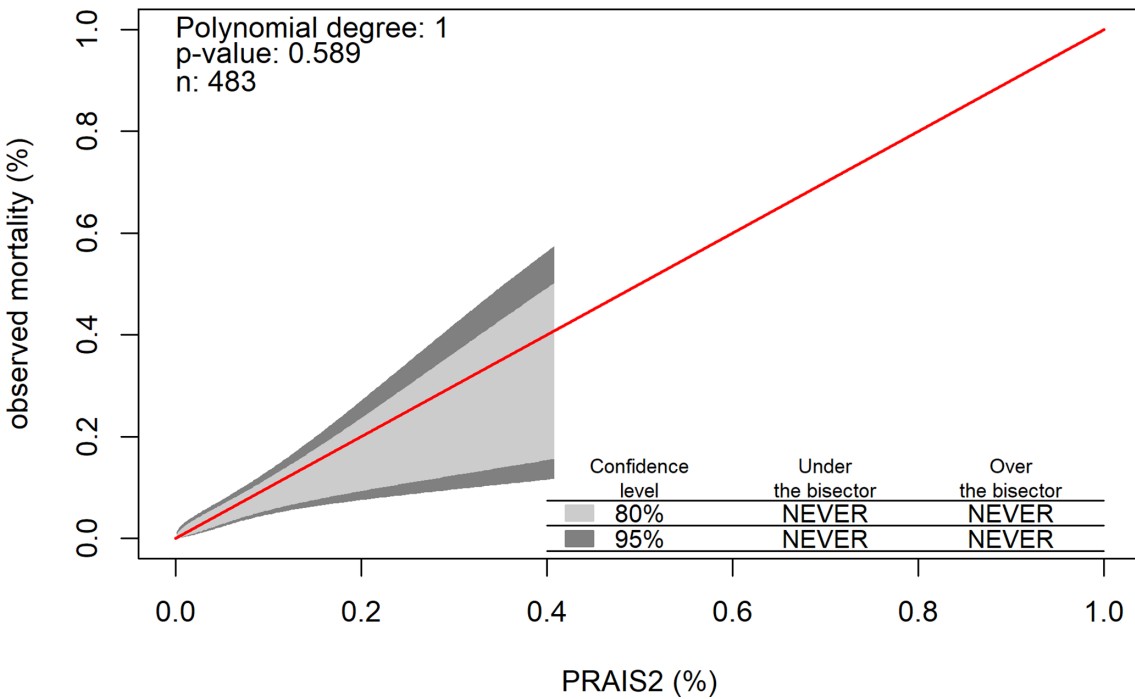

**Figure 4** Calibration belt in the subset of non-elective procedures in the external validation cohort.

(calibration belt) which does not require patients to be categorised but rather compares across the continuous score of risk for our main method of assessing calibration.[29] We also presented Hosmer and Lemeshow goodness of fit test in the supplementary material for comparison with any previous studies.

In conclusion, our study shows good external validity of PRAIS-2 for predicting short-term mortality in paediatric cardiac surgery (the outcome it was developed to predict). We also show preliminary evidence that PRAIS-2 accurately predicts short-term mortality in non-elective procedures. If similar results were found in more diverse external populations, including those outside of the UK, its wider adoption for risk prediction in paediatric patients undergoing cardiac surgery would be appropriate.

**Acknowledgements** We thank Mai Baquedano for support with data management.

**Contributors** LC: Conception and design, data acquisition, analysis and interpretation of data, writing first draft. MC: Critical revision of publication, approval of final publication. RC: Critical revision of publication, approval of final publication. DL: Design, writing first draft, interpretation of results, supervision.

**Funding** This study was supported by the British Heart Foundation Accelerator Award (AA/18/7/34219), which funds LC, and the Bristol National Institute of Health Research Biomedical Research Centre. LC, RC and DAL work in a unit that receives support from the University of Bristol and the UK Medical Research Council (MC_UU_00011/6). DAL is a National Institute of Health Research Senior Investigator (NF-0616-10102).

**Competing interests** DAL has received support from several national and international charity and government grants and from Medtronic Ltd and Roche Diagnostics for research unrelated to that presented here.

**Patient consent for publication** Not required.

**Ethics approval** This study was approved by the institutional review board (IRB) at the Bristol Heart Institute (reference number 250868).

**Provenance and peer review** Not commissioned; externally peer reviewed.

**Data availability statement** Data may be obtained from a third party and are not publicly available. Data belong to University Hospital Bristol NHS Trust and as such as protected by confidentiality. Data sharing request will need to adhere to trust policy on confidentiality.

**ORCID iDs**
Lucia Cocomello http://orcid.org/0000-0002-9967-1861
Rosie Cornish http://orcid.org/0000-0002-2874-7646

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
