## [Reviewer comments · BMJ Open]

ARTICLE DETAILS

TITLE (PROVISIONAL)	EXTERNAL VALIDATION OF THE IMPROVING PARTIAL RISK ADJUSTMENT IN SURGERY (PRAIS2) MODEL FOR 30-DAY MORTALITY AFTER PAEDIATRIC CARDIAC SURGERY
AUTHORS	Cocomello, Lucia; Caputo, Massimo; Cornish, Rosie; Lawlor, Deborah

VERSION 1 – REVIEW

REVIEWER	Roland Matsouaka Duke University
REVIEW RETURNED	04-Jul-2020

GENERAL COMMENTS	Review of the paper External validation of the improving partial risk adjustment in surgery (PRAIS2) model for 30-day mortality after pediatric cardiac surgery Author: Lucia Cocomello et al. Journal: BMJ Open In this paper the authors sought to validate the partial risk adjustment in surgery (PRAIS2) predicting 30-day mortality in patients after a pediatric cardiac surgery. To do so, they leveraged data from the Bristol Royal Hospital for Children in UK with the goal to assess an independent temporal external validation of PRAIS2 and evaluate its generalizability and clinical utility. They considered three Bristol Heart Institute cohorts (2004-2009, 2009-2015, and 2015-2019), with the 2009-2015 being part of the data that served to build the PRAIS2 score. The paper is well written, the statistical analyses are well-conducted, and the results clearly explained. The authors have also highlighted the limitations of this study and suggest further explorations to evaluate PRAIS2 on non-elective procedures. Nevertheless, I have some key points to make to help improve the manuscript: Major points: 1. All the figures and graphs were of poor quality and difficult to read or process. I suggest writing down table directly in the word document and using R ggplot2 package for the Figures 2-4b.
--

	2. While calibration and discrimination are good measures to evaluate a risk-prediction model, the authors would have painted a more complete picture had they also calculated the sensitivity, specificity, positive- and negative-likelihood ratios (and the positive and negative predictive values at specific thresholds (see Usher-Smith et al. [1] and Sanders et al. [2])). I recommend that these measures be evaluated in the context of this paper. Minor points:  1. Title vs. main text on the use of pediatric vs. paediatric: please choose one. 2. Page 15, line 19: The phrase “insufficient death in those undergoing surgical procedures” is a poor word choice as it appears insensitive and cruel. Although I understand the message the authors tried to convey, but the language used in this context is inappropriate. I recommend that the authors find a positive tone explaining the lower mortality rate, while still emphasizing their incapacity to compare the two groups. References:  1. Usher-Smith, J.A., Harshfield, A., Saunders, C.L., Sharp, S.J., Emery, J., Walter, F.M., Muir, K. and Griffin, S.J., 2018. External validation of risk prediction models for incident colorectal cancer using UK Biobank. British journal of cancer, 118(5), pp.750-759. 2. Saunders, C.L., Kilian, B., Thompson, D.J., McGeoch, L.J., Griffin, S.J., Antoniou, A.C., Emery, J.D., Walter, F.M., Dennis, J., Yang, X. and Usher-Smith, J.A., 2020. External validation of risk prediction models incorporating common genetic variants for incident colorectal cancer using UK Biobank. Cancer Prevention Research, 13(6), pp.509-520.
--	--

REVIEWER	Shahzad alam Teerthankar Mahaveer medical college and research centre, India
REVIEW RETURNED	19-Jul-2020

GENERAL COMMENTS	1. It was indeed a nice effort to collect and review the data. Also it is praiseworthy as authors have compared the data of 2 different time period and also analyzed the subgroup (non-elective procedure) which was not done in the original study. 2. ABSTRACT (Page 3 line 41) In the first validation (cohort 1) the model is only marginally calibrated (p-0.051) and the 80% confidence interval falls both under (0.25-0.26) and over (0.00-0.02) the bisector. So as per the calibration belt that there's a tendency to both underestimate and overestimate the risk. Is it right to say only underestimate. 3. RESULTS
--

	 • It would have been better if authors would have included the analysis of non-elective procedure in both the groups. At least the number of non-elective procedure in group 1 deserves a mention. • (Page 13 line 18-22) In cohort 1, although the bisector lies between the 95% Confidence Interval but the p value is only marginally significant. The table should therefore be explained in details regarding its tendency to over/underestimation of risk like it has been done for the cohort from Bristol that contributed to the original PRAIS2. 4. DISCUSSION  • As shown in the study the more recent cohort (cohort 2) performed better for calibration than original Bristol cohort (page 15 line) which the authors attributed partially to improved surgical results during recent years. Is this the only known factor? Could inclusion of non-elective procedure in cohort 2 affect the result? • Forty percent of the procedure in cohort 2 was non-elective and as shown in the study the non-elective procedures performed better in the analysis. So could this be the reason for better performance of cohort 2. This could be determined only if numbers of non-elective procedure in other groups are known. • The cohort 2 performed far better in the analysis than cohort 1 also. The possible reason for the difference is not discussed by the authors.
--	--

REVIEWER	Jeffrey P. Jacobs, MD, FACS, FACC, FCCP Professor of Surgery, University of Florida Congenital Heart Center (HD-308) Division of Thoracic and Cardiovascular Surgery Department of Surgery, University of Florida Gainesville, Florida, United States of America
REVIEW RETURNED	02-Aug-2020

GENERAL COMMENTS	This well written and thoughtful manuscript reports a study that performs an external independent validation of the PRAIS2 score in a cohort from a single tertiary paediatric UK center (in Bristol, South West England). The authors conclude: “In a single centre UK-based cohort, PRAIS2 showed excellent discrimination and calibration in predicting 30-day mortality in paediatric cardiac surgery including in those undergoing non-elective procedures. Our results support a wider adoption of PRAIS2 score in the clinical practice.” The statistical analysis is thorough, and the analysis is appropriate. The last sentence of the text of this paper requires revision. This sentence states: “If similar results were found in more diverse external populations, including those outside of the UK, its use as the standard tool for risk prediction in paediatric patients undergoing cardiac surgery would be recommended” Several other excellent tools for prediction of the risk associated with pediatric and congenital cardiac surgery. For example, please see the following references, which need to be added to the reference section of this paper:
---

1. O'Brien SM, Jacobs JP, Pasquali SK, Gaynor JW, Karamlou T, Welke KF, Filardo G, Han JM, Kim S, Shahian DM, Jacobs ML. The Society of Thoracic Surgeons Congenital Heart Surgery Database Mortality Risk Model: Part 1-Statistical Methodology. *Ann Thorac Surg.* 2015 Sep;100(3):1054-62. doi: 10.1016/j.athoracsur.2015.07.014. Epub 2015 Aug 3. PMID: 26245502.

2. Jacobs JP, O'Brien SM, Pasquali SK, Gaynor JW, Mayer JE Jr, Karamlou T, Welke KF, Filardo G, Han JM, Kim S, Quintessenza JA, Pizarro C, Tchervenkov CI, Lacour-Gayet F, Mavroudis C, Backer CL, Austin EH 3rd, Fraser CD, Tweddell JS, Jonas RA, Edwards FH, Grover FL, Prager RL, Shahian DM, Jacobs ML. The Society of Thoracic Surgeons Congenital Heart Surgery Database Mortality Risk Model: Part 2-Clinical Application. *Ann Thorac Surg.* 2015 Sep;100(3):1063-70. doi: 10.1016/j.athoracsur.2015.07.011. Epub 2015 Aug 3. PMID: 26245504.

3. Jacobs JP, O'Brien SM, Hill KD, Kumar SR, Austin EH 3rd, Gaynor JW, Gruber PJ, Jonas RA, Pasquali SK, Pizarro C, St Louis JD, Meza J, Thibault D, Shahian DM, Mayer JE Jr, Jacobs ML. Refining The Society of Thoracic Surgeons Congenital Heart Surgery Database Mortality Risk Model With Enhanced Risk Adjustment for Chromosomal Abnormalities, Syndromes, and Noncardiac Congenital Anatomic Abnormalities. *Ann Thorac Surg.* 2019 Aug;108(2):558-566. doi: 10.1016/j.athoracsur.2019.01.069. Epub 2019 Mar 7. PMID: 30853592.

The authors might consider adding the above three references to their excellent manuscript in the reference section of this paper.

It is not appropriate to state or recommend that that the PRAIS2 score should become "the standard tool for risk prediction in paediatric patients undergoing cardiac surgery". One can only state that the PRAIS2 score is an excellent tool with excellent discrimination and calibration, and that wider adoption may be appropriate. Other excellent tools also exist, and although PRAIS 2 is excellent, data does not exist to support the recommendation that PRAIS 2 should be THE standard tool for risk prediction in paediatric patients undergoing cardiac surgery.

Please consider adding the above references and revising the above text. Thank you for your consideration.

Below, I have provided several additional suggestions for the authors to consider as they revise this excellent manuscript.

1. The manuscript states:

"These include consensus based methods, such as the risk adjusted classification for congenital heart surgery (RACHS-1) and Aristotle, and more recently empirical research based methods, such as the society of thoracic surgeons-European association of cardiothoracic surgery score (STS-EACTS) and the partial risk adjustment in surgery (PRAIS), which has been developed in the UK and proposed for measuring between centre variation in mortality across the UK."

In the above sentence, please replace: “risk adjusted classification for congenital heart surgery (RACHS-1)” with: “Risk Adjustment for Congenital Heart Surgery-1 Categories (RACHS-1 Categories)”.

Also, in the above sentence, please replace: “the society of thoracic surgeons-European association of cardiothoracic surgery score (STS-EACTS)” with: “The Society of Thoracic Surgeons - European Association for Cardio-Thoracic Surgery Congenital Heart Surgery Mortality Categories (STAT Mortality Categories)”.

The sentence would therefore read:

“These include consensus based methods, such as the Risk Adjustment for Congenital Heart Surgery-1 Categories (RACHS-1 Categories) and Aristotle Score, and more recently empirical research based methods, such as The Society of Thoracic Surgeons - European Association for Cardio-Thoracic Surgery Congenital Heart Surgery Mortality Categories (STAT Mortality Categories) and the partial risk adjustment in surgery (PRAIS), which has been developed in the UK and proposed for measuring between centre variation in mortality across the UK.”

2. The manuscript states:

“This single UK centre contributed to the cohort in which PRAIS2 was originally developed but with data from different time periods than used here, where we explore performance separately in two cohorts: (i) undergoing procedures earlier than those used in PRAIS2 and (ii) undergoing procedures after those used in PRAIS2.”

Please modify this sentence to read:

“This single UK centre contributed to the cohort in which PRAIS2 was originally developed but with data from different time periods than used here, where we explore performance separately in two cohorts: (i) procedures performed earlier than those used in the initial development and validation of PRAIS2 and (ii) procedures performed after those used in the initial development and validation of PRAIS2.

3. The manuscript states:

“Of these 4,885 we had to exclude 575 (12%) for missing data; the remaining 4310 contributed to one of the three temporal cohorts used here.”

What was the mortality in these 575 excluded cases?

What the mortality in these 575 excluded cases different from the mortality in the remaining 4310 cases (stratified by era)?

4. Please provide detailed legends for the Supplementary figures.

	5. In the following 3 sentences, I suggest replacing the word “discovery” with the word “developmental” or “development” as appropriate. “In additional analyses to check that data from Bristol were not notably different to the main UK wide discovery cohort, we showed that the cohort from Bristol that contributed to the original PRAIS2 discovery also had good discrimination, with results similar to the whole PRAIS2 discovery cohort (AUC 0.82 (95%CI 0.77-0.88, Figure 2c).” “We therefore explored the predictive accuracy of PRAIS2 in the Bristol cohort that contributed to the original PRAIS2 discovery and found good discrimination, but the model was only marginally calibrated with a tendency to underestimate risk. It is not uncommon for calibration to differ between cohorts from different regions or over time.” “A key limitation of this study is that the sample size is relatively small and considerably smaller than the cohort used to develop PRAIS2, which included a total of 21838 procedures (combining discovery and validation).” Thus, these 3 sentences would read: “In additional analyses to check that data from Bristol were not notably different to the main UK wide developmental cohort, we showed that the cohort from Bristol that contributed to the original PRAIS2 development also had good discrimination, with results similar to the whole PRAIS2 developmental cohort (AUC 0.82 (95%CI 0.77-0.88, Figure 2c).” “We therefore explored the predictive accuracy of PRAIS2 in the Bristol cohort that contributed to the original PRAIS2 development and found good discrimination, but the model was only marginally calibrated with a tendency to underestimate risk. It is not uncommon for calibration to differ between cohorts from different regions or over time.” “A key limitation of this study is that the sample size is relatively small and considerably smaller than the cohort used to develop PRAIS2, which included a total of 21838 procedures (combining development and validation).”
--	--

VERSION 1 – AUTHOR RESPONSE

Reviewer(s)' Comments to Author:

Reviewer: 1

In this paper the authors sought to validate the partial risk adjustment in surgery (PRAIS2) predicting 30-day mortality in patients after a pediatric cardiac surgery. To do so, they leveraged data from the Bristol Royal Hospital for Children in UK with the goal to assess an independent temporal external validation of PRAIS2 and evaluate its generalizability and clinical utility. They considered three Bristol

Heart Institute cohorts (2004-2009, 2009-2015, and 2015-2019), with the 2009-2015 being part of the data that served to build the PRAIS2 score. The paper is well written, the statistical analyses are well-conducted, and the results clearly explained. The authors have also highlighted the limitations of this study and suggest further explorations to evaluate PRAIS2 on non-elective procedures. Nevertheless, I have some key points to make to help improve the manuscript:

Major points:

1. All the figures and graphs were of poor quality and difficult to read or process. I suggest writing down table directly in the word document and using R ggplot2 package for the Figures 2-4b.

Response: We have now amended figures as required.

METHODS page 7, Figure 1 became Table 1.

FIGURES 2 became Figure 1 done with ggplot2

FIGURES 3,4,5 became FIGURE 2 done with givitiR but with improved resolution

FIGURE 6 became FIGURE 3 done with ggplot2

FIGURE 7 became FIGURE 4 done with givitiR but with improved resolution

2. While calibration and discrimination are good measures to evaluate a risk-prediction model, the authors would have painted a more complete picture had they also calculated the sensitivity, specificity, positive- and negative-likelihood ratios (and the positive and negative predictive values at specific thresholds (see Usher-Smith et al. [1] and Sanders et al. [2])). I recommend that these measures be evaluated in the context of this paper.

Response: We have now performed these additional analyses, with the following added to the paper:

METHODS, statistical analysis, page 10: " Model performance in the main analysis (Cohort 1 and Cohort 2) was also evaluated by reporting sensitivity, specificity, positive- and negative-likelihood ratios (LRP and LRN) and the positive and negative predictive values (PPV and NPV) using cut-off values that included 10%, 20%, 80% and 90% above the cut-off."

RESULTS, page 14: "Sensitivity, Specificity, LRP, LRN, PPV and NPB using different cut-off values of the prediction score are reported in Table 2. Assuming safe effective interventions could reduce those identified at high risk of mortality, in this population we would want to correctly identify the majority of those at risk i.e. minimise false negative results by having a highly specific test, and we would want a positive test to have a high probability of identifying those most at risk (PPV). A 10% threshold maximises specificity (0.90) and maximised the PPV (though this is only 0.10). The 20% threshold also has high specificity (0.80) and a similar PPV (0.06) to the 10% threshold.

Table 2: Metrics of model discrimination according to different PRAIS-2 cut-off values in the Cohort 2

	Metrics	Top 90%	Top 80%	Top 20%	Top 10%
Cohort 2	Cut off	0.001	0.003	0.03	0.07
	Sensitivity	1	1	0.69	0.56
	Specificity	0.10	0.20	0.80	0.90
	PPV	0.02	0.02	0.06	0.10
	NPV	1	1	0.99	0.99
	LRP	1.11	1.25	3.64	6.20
	LRN	0	0	0.37	0.47

LRP= positive--likelihood ratios; LRN= negative-likelihood ratios; PPV= positive predictive values; NPV=negative predictive values “

RESULTS, page 15: “Supplementary Table 7 shows the discrimination metrics for this cohort, with results similar to those found for Cohort 2 and shown in Table 2”

SUPPLEMENTARY Table 7: Metrics of model discrimination according to different PRAIS-2 cut-off values in the Cohort 1

	Metrics	Top 90%	Top 80%	Top 20%	Top 10%
Cohort 1	Cut off	0.0009	0.0025	0.03	0.046
	Sensitivity	1	0.96	0.44	0.19
	Specificity	0.1	0.20	0.80	0.90
	PPV	0.03	0.03	0.05	0.04
	NPV	1	0.99	0.98	0.98
	LRP	1.11	1.21	2.23	1.89
	LRN	0	0.15	0.69	0.90

LRP= positive--likelihood ratios; LRN= negative-likelihood ratios; PPV= positive predictive values; NPV=negative predictive values “

References:

1. Usher-Smith, J.A., Harshfield, A., Saunders, C.L., Sharp, S.J., Emery, J., Walter, F.M., Muir, K. and Griffin, S.J., 2018. External validation of risk prediction models for incident colorectal cancer using UK Biobank. *British journal of cancer*, 118(5), pp.750-759.

2. Saunders, C.L., Kilian, B., Thompson, D.J., McGeoch, L.J., Griffin, S.J., Antoniou, A.C., Emery, J.D., Walter, F.M., Dennis, J., Yang, X. and Usher-Smith, J.A., 2020. External validation of risk prediction models incorporating common genetic variants for incident colorectal cancer using UK Biobank. *Cancer Prevention Research*, 13(6), pp.509-520.

Minor points:

1. Title vs. main text on the use of pediatric vs. paediatric: please choose one.

Response: We have now made sure that paediatric is used throughout the paper”

2. Page 15, line 19: The phrase “insufficient death in those undergoing surgical procedures” is a poor word choice as it appears insensitive and cruel. Although I understand the message the authors tried to convey, but the language used in this context is inappropriate. I recommend that the authors find a positive tone explaining the lower mortality rate, while still emphasizing their incapacity to compare the two groups.

Response: The manuscript amended as suggested:

DISCUSSION, page 17:” We were not able to compare performance between those undergoing elective procedures and those undergoing non-elective procedures as the mortality rate was very low, resulting in only 1 death among those undergoing elective procedures.”

Reviewer: 2

1. It was indeed a nice effort to collect and review the data. Also it is praiseworthy as authors have compared the data of 2 different time period and also analyzed the subgroup (non-elective procedure) which was not done in the original study.

Response: We thank the reviewer for this supportive comment.

2. ABSTRACT (Page 3 line 41)

In the first validation (cohort 1) the model is only marginally calibrated ($p=0.051$) and the 80% confidence interval falls both under (0.25-0.26) and over (0.00-0.02) the bisector. So as per the calibration belt that there’s a tendency to both underestimate and overestimate the risk.

Is it right to say only underestimate.

Response: We agree with the reviewer's observation and have amended the description of these results in the abstract:

ABSTRACT, page 2:" While PRAIS-2 was only marginally calibrated in Cohort 1, with a tendency to underestimate risk in low risk and overestimate risk in high risk procedures (P-value = 0.033), validation in Cohort 2 showed good calibration with the 95% confidence belt containing the bisector for predicted mortality (P-value = 0.143)."

We have also made sure that they are correctly described in the results section of the paper

3. RESULTS

- It would have been better if authors would have included the analysis of non-elective procedure in both the groups. At least the number of non-elective procedure in group 1 deserves a mention.

Response: This information was not recorded in the initial cohort and therefore we could not stratify the analysis for non-elective vs elective procedures in this subgroup. We have edited the revised paper to now make this clearer:

RESULTS, page 15:" In the subgroup of non-elective procedures (a subgroup of Cohort 2 as these data were not available in the other two cohorts)"

- (Page 13 line 18-22) In cohort 1, although the bisector lies between the 95% Confidence Interval but the p value is only marginally significant. The table should therefore be explained in details regarding its tendency to over/underestimation of risk like it has been done for the cohort from Bristol that contributed to the original PRAIS2.

Response; The description of these results has been updated accordingly:

RESULTS, page 15: " In Cohort 1, PRAIS-2 showed a lower but still good discrimination ability (AUC 0.72 (95%CI: 0.65 to 0.80, Figure 1), compared with Cohort 1, but was poorly calibrated, with a tendency to underestimate risk in low risk procedures and overestimate in high risk procedures (p-value=0.033, Figure 2."

4. DISCUSSION

- As shown in the study the more recent cohort (cohort 2) performed better for calibration than original Bristol cohort (page 15 line) which the authors attributed partially to improved surgical results during recent years. Is this the only known factor? Could inclusion of non-elective procedure in cohort 2 affect the result?
- Forty percent of the procedure in cohort 2 was non-elective and as shown in the study the non-elective procedures performed better in the analysis. So could this be the reason for better performance of cohort 2. This could be determined only if numbers of non-elective procedure in other groups are known.

- The cohort 2 performed far better in the analysis than cohort 1 also. The possible reason for the difference is not discussed by the authors.

Response: As noted above whether the procedure was elective or non-elective has only recently been recorded in these data and so is not available in either Cohort 1 or the Overlapping cohort.

We speculate that tendency to miscalibration in the earliest cohort and good calibration in the most recent cohorts, is likely to reflect changes over time in case mix and patient risk profile but also improvement in perioperative care. The good calibration in the most recent cohort also suggests that recent local results were consistent with those reported nationally.

We have added this sentences in discussion, page 16:

“We speculate that the temporal improvement in calibration across all three cohorts from the same geographical region is likely to reflect changes over time in case mix, patient risk profile and improvement in perioperative care. Whist we saw an improvement in calibration over time the amount of this was modest, with the Hosmer-Lemeshow test X2 going from 19 in Cohort 1, through 16 in the Overlapping Cohort to 8 in Cohort 2.”

Reviewer: 3

The authors conclude:

“In a single centre UK-based cohort, PRAIS2 showed excellent discrimination and calibration in predicting 30-day mortality in paediatric cardiac surgery including in those undergoing non-elective procedures. Our results support a wider adoption of PRAIS2 score in the clinical practice.”

The statistical analysis is thorough, and the analysis is appropriate.

The last sentence of the text of this paper requires revision.

This sentence states:

“If similar results were found in more diverse external populations, including those outside of the UK, its use as the standard tool for risk prediction in paediatric patients undergoing cardiac surgery would be recommended”

Several other excellent tools for prediction of the risk associated with pediatric and congenital cardiac surgery. For example, please see the following references, which need to be added to the reference section of this paper:

1. O'Brien SM, Jacobs JP, Pasquali SK, Gaynor JW, Karamlou T, Welke KF, Filardo G, Han JM, Kim S, Shahian DM, Jacobs ML. The Society of Thoracic Surgeons Congenital Heart Surgery Database Mortality Risk Model: Part 1-Statistical Methodology. *Ann Thorac Surg.* 2015 Sep;100(3):1054-62. doi: 10.1016/j.athoracsur.2015.07.014. Epub 2015 Aug 3. PMID: 26245502.
2. Jacobs JP, O'Brien SM, Pasquali SK, Gaynor JW, Mayer JE Jr, Karamlou T, Welke KF, Filardo G, Han JM, Kim S, Quintessenza JA, Pizarro C, Tchervenkov CI, Lacour-Gayet F, Mavroudis C, Backer CL, Austin EH 3rd, Fraser CD, Tweddell JS, Jonas RA, Edwards FH, Grover FL, Prager RL, Shahian DM, Jacobs ML. The Society of Thoracic Surgeons Congenital Heart Surgery Database Mortality Risk Model: Part 2-Clinical Application. *Ann Thorac Surg.* 2015 Sep;100(3):1063-70. doi: 10.1016/j.athoracsur.2015.07.011. Epub 2015 Aug 3. PMID: 26245504.
3. Jacobs JP, O'Brien SM, Hill KD, Kumar SR, Austin EH 3rd, Gaynor JW, Gruber PJ, Jonas RA, Pasquali SK, Pizarro C, St Louis JD, Meza J, Thibault D, Shahian DM, Mayer JE Jr, Jacobs ML. Refining The Society of Thoracic Surgeons Congenital Heart Surgery Database Mortality Risk Model With Enhanced Risk Adjustment for Chromosomal Abnormalities, Syndromes, and Noncardiac Congenital Anatomic Abnormalities. *Ann Thorac Surg.* 2019 Aug;108(2):558-566. doi: 10.1016/j.athoracsur.2019.01.069. Epub 2019 Mar 7. PMID: 30853592.

The authors might consider adding the above three references to their excellent manuscript in the reference section of this paper.

It is not appropriate to state or recommend that that the PRAIS2 score should become “the standard tool for risk prediction in paediatric patients undergoing cardiac surgery”. One can only state that the PRAIS2 score is an excellent tool with excellent discrimination and calibration, and that wider adoption may be appropriate. Other excellent tools also exist, and although PRAIS 2 is excellent, data does not exist to support the recommendation that PRAIS 2 should be THE standard tool for risk prediction in paediatric patients undergoing cardiac surgery.

Response: Our original conclusion was based on the fact that PRAIS-2 was developed specifically for use in the UK and is the tool that we have assessed. However, we agree that the changes suggested by the reviewer are pertinent to a publication in an international journal. We have amended the concluding sentence to address this

DISCUSSION, page 18:” If similar results were found in more diverse external populations, including those outside of the UK, its wider adoption for risk prediction in paediatric patients undergoing cardiac surgery would be appropriate.”

We have also made reference to the other prediction tools highlight by the reviewer in the

Below, I have provided several additional suggestions for the authors to consider as they revise this excellent manuscript.

1. The manuscript states:

“These include consensus based methods, such as the risk adjusted classification for congenital heart surgery (RACHS-1) and Aristotle, and more recently empirical research based methods, such as the society of thoracic surgeons-European association of cardiothoracic surgery score (STS-EACTS) and the partial risk adjustment in surgery (PRAIS), which has been developed in the UK and proposed for measuring between centre variation in mortality across the UK.”

In the above sentence, please replace: “risk adjusted classification for congenital heart surgery (RACHS-1)” with: “Risk Adjustment for Congenital Heart Surgery-1 Categories (RACHS-1 Categories)”.

The sentence would therefore read:

“These include consensus based methods, such as the Risk Adjustment for Congenital Heart Surgery-1 Categories (RACHS-1 Categories) and Aristotle Score, and more recently empirical research based methods, such as The Society of Thoracic Surgeons - European Association for Cardio-Thoracic Surgery Congenital Heart Surgery Mortality Categories (STAT Mortality Categories) and the partial risk adjustment in surgery (PRAIS), which has been developed in the UK and proposed for measuring between centre variation in mortality across the UK.”

Response: We have changed the sentence exactly as suggested by the reviewer

2. The manuscript states:

“This single UK centre contributed to the cohort in which PRAIS2 was originally developed but with data from different time periods than used here, where we explore performance separately in two cohorts: (i) undergoing procedures earlier than those used in PRAIS2 and (ii) undergoing procedures after those used in PRAIS2.”

Please modify this sentence to read:

“This single UK centre contributed to the cohort in which PRAIS2 was originally developed but with data from different time periods than used here, where we explore performance separately in two cohorts: (i) procedures performed earlier than those used in the initial development and validation of PRAIS2 and (ii) procedures performed after those used in the initial development and validation of PRAIS2.

Response: Manuscript amended as suggested

INTRODUCTION, page 5:” This single UK centre contributed to the cohort in which PRAIS-2 was originally developed, but with data from different time periods than used here. In this study we determine performance discrimination separately in two Bristol Heart Institute cohorts: (i) procedures performed earlier than those used in the initial development and validation of PRAIS-2 (Cohort 1) and (ii) procedures performed after those used in the initial development and validation of PRAIS-2 (Cohort 2).”

3. The manuscript states:

“Of these 4,885 we had to exclude 575 (12%) for missing data; the remaining 4310 contributed to one of the three temporal cohorts used here.”

What was the mortality in these 575 excluded cases? What the mortality in these 575 excluded cases different from the mortality in the remaining 4310 cases (stratified by era)?

Response: These results are now presented in the revised manuscript and compared with mortality in the group included in analyses:

METHOD, Data source, page 8

“The entire (i.e. including all 3 cohorts used in any analyses) Bristol data consisted of 4,886 paediatric cardiac surgical procedures (defined as surgery on the heart or great vessel in patients aged < 16 years old, excluding catheter procedures and trivial/minor procedures) performed between April 2004 and July 2019. Patients with missing information on mortality (133), one or more of the variables used in the calculation of PRAIS-2 calculation (371) or both (9) were excluded. Patients with missing information on one or more of the variables used to calculate PRAIS-2 showed a higher rate of 30-day mortality (Supplementary table 1). For those with missing data on one or more of the variables used to calculate PRAIS-2, the variables that were available suggested that they had higher risk profiles than those with no missing variables, both in the whole study data (Supplementary table 1) and when analysed in the three separate temporal cohorts (Supplementary table 2-4). For example, in those with at least one missing variable (N = 371) the proportion with severe illness was 22.9%, whereas in

those with no missing variables (N = 4373) it was 7%. The same results in cohort 1 were 15.5% and 0%, in the overlapping with PRAIS-2 discovery were 0% and 3.3% and in cohort 2 were 18.5% and 55.6%. The remaining 4373 (90% of the 4886 eligible) procedures were included in the analysis.”

DISCUSSION, page 17:” Further, 10.5% of procedures had missing data on mortality and/or variables used to calculate PRAIS2. These individuals tended to have higher risk profiles and their mortality rates were higher.”

Supplementary table 1-4:

Supplementary Table 1 Procedures with missing data on PRAIS2 compared with procedures with complete data

	Non-missing	PRAIS2 Missing
n	4373	371
Age (median, IQR)	2.55(4.02)	3.39(4.74)
Diagnoses group n (%)^a		
GROUP 1	332(7.6)	51(13.7)
GROUP 2	506(11.6)	46(12.4)
GROUP 3	339(7.8)	27(7.3)
GROUP 4	672(15.4)	33(8.9)
GROUP 5	349(8.0)	36(9.7)
GROUP 6	341(7.8)	52(14.0)
GROUP 7	615(14.1)	21(5.7)
GROUP 8	452(10.3)	18(4.9)
GROUP 9	47(1.1)	1(0.3)
GROUP 10	165(3.8)	6(1.6)
GROUP 11	555(12.7)	21(5.7)
GROUP NA	0(0.0)	59(15.9)
Procedure Group n (%)^a		
GROUP 1	60(1.4)	0(0.0)

GROUP 2	73(1.7)	0(0.0)
GROUP 3	237(5.4)	0(0.0)
GROUP 4	316(7.2)	13(3.5)
GROUP 5	811(18.5)	7(1.9)
GROUP 6	381(8.7)	5(1.3)
GROUP 7	116(2.7)	2(0.5)
GROUP 8	312(7.1)	3(0.8)
GROUP 9	62(1.4)	0(0.0)
GROUP 10	141(3.2)	0(0.0)
GROUP 11	112(2.6)	1(0.3)
GROUP 12	146(3.3)	0(0.0)
GROUP 13	258(5.9)	1(0.3)
GROUP 14	136(3.1)	2(0.5)
GROUP 15	636(14.5)	5(1.3)
GROUP 20	576(13.2)	9(2.4)
GROUP NA	0(0.0)	323(87.1)
Bypass n (%)	3175(72.6)	51(13.7)
Weight (mean, SD)	11.54(13.86)	14.16(16.78)
UVH category, n (%)		
No	3753(85.8)	274(73.9)
Yes	620(14.2)	73(19.7)
NA	0(0.0)	24(6.5)
Severity of illness^b, n (%)		
No	4065(93.0)	271(73.0)
Yes	308(7.0)	85(22.9)

NA	0(0.0)	15(4.0)
Acquired comorbidity^b, n (%)		
No	4063(92.9)	299(80.6)
Yes	310(7.1)	57(15.4)
NA	0(0.0)	15(4.0)
Additional cardiac risk factors^b, n (%)		
No	4181(95.6)	324(87.3)
Yes	192(4.4)	32(8.6)
NA	0(0.0)	15(4.0)
Congenital comorbidity^b, n (%)		
No	3583(81.9)	280(75.5)
Yes	790(18.1)	76(20.5)
Na	0(0.0)	15(4.0)
30-day mortality, n (%)	111(2.5)	31(8.4)

^a as the diagnoses groups and procedure groups have lengthy text details, we have not provided them in the table; they can be found in Supplementary Table 5.

^b Definitions of variables are given in Supplementary Table 5

Supplementary table 2 Procedures with missing data on PRAIS2 compared with procedures with complete data in the Cohort 1

	Non-missing	Missing
n	1352	58

Age (median, IQR)	2.39(3.93)	3.59(4.84)
Diagnoses group n (%) ^a		
GROUP 1	71(5.3)	2(3.4)
GROUP 2	149(11.0)	5(8.6)
GROUP 3	111(8.2)	5(8.6)
GROUP 4	218(16.1)	5(8.6)
GROUP 5	104(7.7)	6(10.3)
GROUP 6	104(7.7)	9(15.5)
GROUP 7	176(13.0)	1(1.7)
GROUP 8	155(11.5)	3(5.2)
GROUP 9	20(1.5)	0(0.0)
GROUP 10	56(4.1)	1(1.7)
GROUP 11	188(13.9)	2(3.4)
GROUP NA	0(0.0)	19(32.8)
Procedure Group n (%) ^a		
GROUP 1	1(0.1)	0(0.0)
GROUP 2	29(2.1)	0(0.0)
GROUP 3	107(7.9)	0(0.0)
GROUP 4	99(7.3)	7(12.1)
GROUP 5	239(17.7)	1(1.7)
GROUP 6	148(10.9)	2(3.4)
GROUP 7	21(1.6)	1(1.7)
GROUP 8	86(6.4)	0(0.0)
GROUP 9	20(1.5)	0(0.0)
GROUP 10	36(2.7)	0(0.0)

GROUP 11	36(2.7)	1(1.7)
GROUP 12	32(2.4)	0(0.0)
GROUP 13	87(6.4)	1(1.7)
GROUP 14	38(2.8)	0(0.0)
GROUP 15	217(16.1)	3(5.2)
GROUP 20	156(11.5)	4(6.9)
GROUP NA	0(0.0)	38(65.5)
Bypass n (%)	903(66.8)	19(32.8)
Weight (mean, SD)	11.04(13.73)	14.34(16.21)
UVH category, n (%)		
No	1209(89.4)	48(82.8)
Yes	143(10.6)	3(5.2)
NA	0(0.0)	7(12.1)
Severity of illness^b, n (%)		
NA	0(0.0)	9(15.5)
Acquired comorbidity^b, n (%)		
No	1343(99.3)	49(84.5)
Yes	9(0.7)	0(0.0)
NA	0(0.0)	9(15.5)
Additional cardiac risk factors^b, n (%)		
No	1294(95.7)	45(77.6)
Yes	58(4.3)	4(6.9)
NA	0(0.0)	9(15.5)
Congenital comorbidity^b, n (%)		
No	1179(87.2)	42(72.4)

Yes	173(12.8)	7(12.1)
NA	0(0.0)	9(15.5)
30-day mortality, n (%)	32(2.4)	2(3.4)

^a as the diagnoses groups and procedure groups have lengthy text details, we have not provided them in the table; they can be found in Supplementary Table 5

^b Definitions of variables are given in Supplementary Table 5

Supplementary table 3 Procedures with missing data on PRAIS2 compared with procedures with complete data in the Cohort 2

	Non-missing	Missing
n	1197	133
Age (mean, SD)	2.91(4.22)	3.49(4.76)
Diagnoses group n (%)^a		
GROUP 1	107(8.9)	25(18.8)
GROUP 2	163(13.6)	12(9.0)
GROUP 3	103(8.6)	14(10.5)
GROUP 4	232(19.4)	16(12.0)
GROUP 5	122(10.2)	23(17.3)
GROUP 6	137(11.4)	26(19.5)
GROUP 7	127(10.6)	4(3.0)
GROUP 8	105(8.8)	3(2.3)
GROUP 9	2(0.2)	0(0.0)
GROUP 10	14(1.2)	1(0.8)
GROUP 11	85(7.1)	5(3.8)
GROUP NA	0(0.0)	4(3.0)

Procedure Group n (%)^a		
GROUP 1	31(2.6)	0(0.0)
GROUP 2	11(0.9)	0(0.0)
GROUP 3	46(3.8)	0(0.0)
GROUP 4	103(8.6)	0(0.0)
GROUP 5	228(19.0)	0(0.0)
GROUP 6	53(4.4)	0(0.0)
GROUP 7	45(3.8)	0(0.0)
GROUP 8	100(8.4)	0(0.0)
GROUP 9	16(1.3)	0(0.0)
GROUP 10	35(2.9)	0(0.0)
GROUP 11	19(1.6)	0(0.0)
GROUP 12	64(5.3)	0(0.0)
GROUP 13	68(5.7)	0(0.0)
GROUP 14	43(3.6)	0(0.0)
GROUP 15	144(12.0)	0(0.0)
GROUP 20	191(16.0)	0(0.0)
GROUP NA	0(0.0)	133(100.0)
Bypass n (%)	911(76.1)	0(0.0)
Weight (mean, SD)	12.57(13.89)	14.31(15.94)
UVH category, n (%)		
GROUP 0	978(81.7)	98(73.7)
GROUP 1.2	219(18.3)	31(23.3)
GROUP NA.2	0(0.0)	4(3.0)
Severity of illness^b, n (%)		

Yes	222(18.5)	74(55.6)
Acquired comorbidity^b, n (%)		
Yes	207(17.3)	43(32.3)
Additional cardiac risk factors^b, n (%)		
yes	82(6.9)	17(12.8)
Congenital comorbidity^b, n (%)	343(28.7)	41(30.8)
30-day mortality, n (%)	23(1.9)	14(10.5)

^a as the diagnoses groups and procedure groups have lengthy text details, we have not provided them in the table; they can be found in Supplementary Table 5.

^b Definitions of variables are given in Supplementary Table 5

Supplementary table 4 Procedures with missing data on PRAIS2 compared with procedures with complete data in the Overlapping Cohort

	Non-missing	Missing
n	1824	180
Age (mean, SD)	2.43(3.93)	3.26(4.71)
Diagnoses group n (%)^a		
GROUP 1	154(8.4)	24(13.3)
GROUP 2	194(10.6)	29(16.1)
GROUP 3	125(6.9)	8(4.4)
GROUP 4	222(12.2)	12(6.7)
GROUP 5	123(6.7)	7(3.9)
GROUP 6	100(5.5)	17(9.4)
GROUP 7	312(17.1)	16(8.9)
GROUP 8	192(10.5)	12(6.7)

GROUP 9	25(1.4)	1(0.6)
GROUP 10	95(5.2)	4(2.2)
GROUP 11	282(15.5)	14(7.8)
GROUP NA	0(0.0)	36(20.0)
Procedure Group n (%)^a		
GROUP 1	28(1.5)	0(0.0)
GROUP 2	33(1.8)	0(0.0)
GROUP 3	84(4.6)	0(0.0)
GROUP 4	114(6.2)	6(3.3)
GROUP 5	344(18.9)	6(3.3)
GROUP 6	180(9.9)	3(1.7)
GROUP 7	50(2.7)	1(0.6)
GROUP 8	126(6.9)	3(1.7)
GROUP 9	26(1.4)	0(0.0)
GROUP 10	70(3.8)	0(0.0)
GROUP 11	57(3.1)	0(0.0)
GROUP 12	50(2.7)	0(0.0)
GROUP 13	103(5.6)	0(0.0)
GROUP 14	55(3.0)	2(1.1)
GROUP 15	275(15.1)	2(1.1)
GROUP 20	229(12.6)	5(2.8)
GROUP NA	0(0.0)	152(84.4)
Bypass n (%)	1361(74.6)	32(17.8)
Weight (mean, SD)	11.23(13.92)	13.99(17.63)
UVH category, n (%)		

No	1566(85.9)	128(71.1)
Yes	258(14.1)	39(21.7)
NA	0(0.0)	13(7.2)
Severity of illness^b, n (%)		
No	1738(95.3)	163(90.6)
Yes	86(4.7)	11(6.1)
NA	0(0.0)	6(3.3)
Acquired comorbidity^b, n (%)		
No	1730(94.8)	160(88.9)
Yes	94(5.2)	14(7.8)
NA	0(0.0)	6(3.3)
Additional cardiac risk factors^b, n (%)		
No	1772(97.1)	163(90.6)
Yes	52(2.9)	11(6.1)
NA	0(0.0)	6(3.3)
Congenital comorbidity^b, n (%)		
No	1550(85.0)	146(81.1)
Yes	274(15.0)	28(15.6)
NA	0(0.0)	6(3.3)
30-day mortality, n (%)	56(3.1)	15(8.3)

^a as the diagnoses groups and procedure groups have lengthy text details, we have not provided them in the table; they can be found in Supplementary Table 5.

^b Definitions of variables are given in Supplementary Table 5

4. Please provide detailed legends for the Supplementary figures.

Response: We have modified the figure legend:” Sinaplot of PRAIS2 distribution stratified for incidence of 30-day mortality, 0=survivors and 1= not survivors”

5. In the following 3 sentences, I suggest replacing the word “discovery” with the word “developmental” or “development” as appropriate.

“In additional analyses to check that data from Bristol were not notably different to the main UK wide discovery cohort, we showed that the cohort from Bristol that contributed to the original PRAIS2 discovery also had good discrimination, with results similar to the whole PRAIS2 discovery cohort (AUC 0.82 (95%CI 0.77-0.88, Figure 2c).”

“We therefore explored the predictive accuracy of PRAIS2 in the Bristol cohort that contributed to the original PRAIS2 discovery and found good discrimination, but the model was only marginally calibrated with a tendency to underestimate risk. It is not uncommon for calibration to differ between cohorts from different regions or over time.”

“A key limitation of this study is that the sample size is relatively small and considerably smaller than the cohort used to develop PRAIS2, which included a total of 21838 procedures (combining discovery and validation).”

Response: we have amended this sentence as suggested

uncommon for calibration to differ between cohorts from different regions or over time.”

VERSION 2 – REVIEW

REVIEWER	Roland Matsouaka Duke University
REVIEW RETURNED	19-Oct-2020

GENERAL COMMENTS	I thank the authors for such a quick turnover in getting back to us and, most importantly, for the work well done. The authors have been thorough in their response to the reviewers’ comments. They have been very responsive in their responsive to the suggestions made by the reviewers, making the necessary changes in the manuscript and providing clear details to some of the key points made. I don’t have further comments and recommend that the paper be accepted for publication.
--

REVIEWER	DR SHAHZAD ALAM
-----------------	-----------------

	TEERTHANKAR MAHAVEER MEDICAL COLLEGE AND RESEARCH CENTER, INDIA
REVIEW RETURNED	21-Oct-2020

GENERAL COMMENTS	The correction made to the manuscript are satisfactory.
---

REVIEWER	Jeffrey Phillip Jacobs, MD The University of Florida United States of America
REVIEW RETURNED	31-Oct-2020

GENERAL COMMENTS	I previously reviewed this excellent paper on July 31, 20202. I congratulate and thank the authors for being responsive to the Reviewers. I have no additional questions or queries. This paper is now ready for publication. This well written and thoughtful manuscript reports a study that performs an external independent validation of the PRAIS2 score in a cohort from a single tertiary paediatric UK center (in Bristol, South West England). The authors conclude: “In a single centre UK-based cohort, PRAIS-2 showed excellent discrimination and calibration in predicting 30-day mortality in paediatric cardiac surgery including in those undergoing non-elective procedures. Our results support a wider adoption of PRAIS-2 score in the clinical practice.” The statistical analysis is thorough, and the analysis is appropriate. The Discussion is thoughtful and the Conclusions are appropriate.
--